# Lithium Salt Catalyzed Ring-Opening Polymerized Solid-State Electrolyte with Comparable Ionic Conductivity and Better Interface Compatibility for Li-Ion Batteries

**DOI:** 10.3390/membranes12030330

**Published:** 2022-03-16

**Authors:** Wei Zhang, Sujin Yoon, Lei Jin, Hyunmin Lim, Minhyuk Jeon, Hohyoun Jang, Faiz Ahmed, Whangi Kim

**Affiliations:** 1Department of Applied Chemistry, Konkuk University, Chungju 27478, Korea; arno_zw@hotmail.com (W.Z.); ysj920126@naver.com (S.Y.); jinlei8761@naver.com (L.J.); tree3367@naver.com (H.L.); jeonminh97@naver.com (M.J.); 201417450@kku.ac.kr (H.J.); 2Grenoble INP, LEPMI, University of Grenoble Alpes, 38000 Grenoble, France; faiz2310@gmail.com

**Keywords:** in situ polymerization, interfacial issues, LiFSI, ionic conductivity, Li-ion battery

## Abstract

Rechargeable lithium-ion batteries have drawn extensive attention owing to increasing demands in applications from portable electronic devices to energy storage systems. In situ polymerization is considered one of the most promising approaches for enabling interfacial issues and improving compatibility between electrolytes and electrodes in batteries. Herein, we observed in situ thermally induced electrolytes based on an oxetane group with LiFSI as an initiator, and investigated structural characteristics, physicochemical properties, contacting interface, and electrochemical performances of as-prepared SPEs with a variety of technologies, such as FTIR, ^1^H-NMR, FE-SEM, EIS, LSV, and chronoamperometry. The as-prepared SPEs exhibited good thermal stability (stable up to 210 °C), lower activation energy, and high ionic conductivity (>0.1 mS/cm) at 30 °C. Specifically, SPE-2.5 displayed a comparable ionic conductivity (1.3 mS/cm at 80 °C), better interfacial compatibility, and a high Li-ion transference number. The SPE-2.5 electrolyte had comparable coulombic efficiency with a half-cell configuration at 0.1 C for 50 cycles. Obtained results could provide the possibility of high ionic conductivity and good compatibility through in situ polymerization for the development of Li-ion batteries.

## 1. Introduction

Rechargeable Li-ion batteries (LiBs) have acted as an essential part of our daily lives in electronic items ranging from portable consumer electronic devices to large-scale transport equipment such as electric vehicles [1]. To date, conventional organic or inorganic liquid electrolytes (LEs) have some intrinsic or extrinsic properties (e.g., flammability, cost-inefficiency, and low ionic conductivity) that hinder extensive applications of current LiBs in energy storage and conversion systems. Extensive concern has been expressed about GPEs and SPEs due to their safety, thermal stability, and high voltage window [2]. Nevertheless, SPEs that can outperform LEs remain a challenge because of interfacial issues and compatibility between solid electrolytes and electrodes [3,4,5], which hinder Li^+^ transportation and impair the cycling performance of batteries. Thus, several strategies have been studied (e.g., thermally induced, free-radical, ionic, and ultraviolet (UV) in situ polymerization) for polymer electrolytes [6]. UV and thermally induced free-radical polymerization techniques still require development before they can be applied in industry.

In situ thermally induced CROP is considered one of the most promising approaches, with the advantages of effectively reducing interface resistance and enabling the compatibility of commercial components of LiBs production [2,3], which can produce the expected polymers for the development of SPEs. Several industrial polymers have been produced through the CROP technique, such as polytetrahydrofurans, polysiloxanes, polyoxymethylene, and polyethyleneimines [7]. The CROP reaction can be initiated by Brønsted acids (e.g., HCl, H_2_SO_4_, HClO_4_, and HOSO_2_CF_3_) and Lewis acids with co-initiators, photoinitiators, and onium ions [8]. BF_3_ is one of the widely employed Lewis acids for CROP [2,3], as is its precursor boron trifluoride diethyl ether complex (BF_3_ OEt_2_) [9]. Moreover, common lithium salts such as lithium tetrafluoroborate (LiBF_4_) [10], lithium perchlorate (LiClO_4_) [2,9], lithium hexafluorophosphate (LiPF_6_) [10], lithium difluoro(oxalato)borate (LiDFOB) [11], and lithium bis(trifluoromethane)sulfonimide (LiTFSI) [12] have been introduced into ring-opening polymerization because of their superacid anions. Lithium salts can also act as Lewis acids to activate the ring-opening polymerization reaction owing to their lower melting point and better solubility in low dielectric media [13]. For example, LiBF_4_ can be treated like fluorine compounds and the Lewis acid BF_3_, which has also been regarded as a superacid anion. Notably, aluminum triflate Al(CF_3_SO_3_)_3_ and Al(OTf)_3_ salts are efficient initiators for the polymerization of DOL [14]. The produced SPEs exhibit high ionic conductivity at ambient temperature, low interfacial resistance, highly coulombic efficiency (>99%), and long lifespan through in situ-formed SPEs. Some researchers have also studied polymerization based on the oxetane group using lithium salts as initiators, including LiBF_4_ [15], LiPF_6_ [10], and LiN(C_2_F_5_SO_2_)_2_ [9] with solvents. However, the poor conductivity of the obtained SPEs could be attributed to the incorporation of plasticizer or organic solvent (e.g., acetone, THF, acetonitrile) with inert fragility [2,13]. No additional solvents which take part in polymerized electrolytes are proposed. Compared with commercial LiPF_6_, LiFSI has been extensively studied as a promising alternative conducting salt for LiBs, which not only exhibit superior stability and higher σ [16], but can also enhance electrochemical cyclability with graphite or Li metal anode [17,18]. Our group successfully fabricated siloxane-epoxy polymerized electrolytes (SEPEs) through in situ cationic ring opening, which illustrated low interfacial resistance, high conductivity (0.116 mS/cm), and a lithium transference number (*t*_Li+_) of 0.61 at room temperature. Moreover, the SEPEs also demonstrate a wide electrochemical stability window (up to ca. 4.7 V vs. Li/Li^+^) [19]. However, synthesized SEPEs with organic solvent could reduce battery safety [20]. Researchers have also studied polymer electrolytes with a trimethylene oxide (TMO) structure through the ring-opening polymerization of oxetane derivatives via lithium salts as an initiator [15,21]. However, the potential flammability and low ionic conductivity of poly(oxetane)-based electrolytes remain challenges in the application of lithium batteries.

Herein, we present a polymer electrolyte based on the oxetane-ring group, utilizing LiFSI salts as an initiator through in situ thermally induced CROP. LiFSI acted as H^+^ capturer, which was utilized to directly self-catalyze the CROP of EOM with an oxetane ring and a hydroxy group, in the absence of organic solvents. The prepared SPEs exhibited good thermostability, better interfacial compatibility among components in the cell, and comparable conductivity (>0.1 mS/cm). We hope this work provides a simplified and efficient assembly strategy for the development of solid-state LiBs.

## 2. Experimental

### 2.1. Chemicals

EOM (96%) was purchased from TCI company (Tokyo, Japan). LiFePO_4_ (LFP)-coated Al foil and lithium foil were obtained from MTI corporation (Richmond, CA, USA). High-purity LiFSI (99.9%), DMSO-*d*_6_, LiCoO_2_ (LCO), poly (vinylidene difluoride) (PVDF), and carbon black (CB) were obtained from Sigma Aldrich (St. Louis, MO, USA), and were used as received unless stated otherwise. Besides, there is a list of abbreviation nomenclature for this article in Appendix A.

### 2.2. Instruments and Measurements

The structural characteristics of as-synthesized electrolytes were analyzed by FTIR spectra (Nicolet iS5, ASB1100426, Thermo Fisher Scientific, Waltham, MA, USA) and ^1^H-NMR (Avance 400FT-NMR (400 Hz), Bruker DRX, Seongnam Korea) with tetramethyl silane (TMS) as a reference. DMSO-*d*_6_ was used as a solvent. Thermal properties were recorded with a Scinco TGA-N 1000 (Seoul, Korea) analyzer from 30 °C to 600 °C at a heating rate of 10 °C/min in the presence of N_2_ atmosphere. The viscosities of electrolytes were observed with a viscometer (*hts*-VROCTM, RheoSense, San Ramon, CA, USA). The morphology of the electrolytes and electrodes inside the cell was attained with a field-emission scanning electron microscope (FE-SEM, JEOL 7401 F, JEOL, Tokyo, Japan).

### 2.3. Fabrication of Asymmetry Dummy Cell

An asymmetric dummy cell was assembled for evaluating the conductivity of the as-synthesized electrolytes, and its configuration is shown in Figure 1. Typically, for cathode electrodes, the composition of slurry paste was LCO (86 wt%), CB (5 wt%), and PVDF (9 wt%). The LiCoO_2_-coated electrode was prepared by doctor-blading a home-made paste on a clean FTO glass electrode (active area: ca. 0.138 cm^2^; thickness: ca. 10 μm) and dried at 80 °C overnight in a vacuum oven. Similarly, a graphite anode electrode was also fabricated on a cleaned FTO glass electrode (active area: ca. 0.138 cm^2^; thickness: ca.12 μm) using a homemade paste consisting of carbon black and PVDF of 90 and 10 wt%, respectively.

### 2.4. In Situ Thermal Polymerization

Cationic ring-opening polymerization of EOM was initiated directly with LiFSI salt. The precursor EOM (1 mL, 8.78 mmol) and LiFSI salt (376 mg, 2.01 mmol) were mixed in 5 mL bottle under a N_2_-filled glove box and the solution was stirred for 2 h to make a homogeneous solution. Afterward, the homogeneous liquid solution was placed in an environment with a constant temperature of 60 °C for 48 h. To evaluate ionic conductivity, freshly prepared homogeneous liquid solution was injected into the dummy cell through the pre-drilled hole on the FTO glass electrode, sealed using Scotch tape, and kept in the same environment. The as-prepared electrolyte of 2 M LiFSI in EOM was designated as SPE-2. Similarly, the electrolytes of 2.5 M LiFSI and 3 M LiFSI in EOM were denoted as SPE-2.5 and SPE-3, respectively.

### 2.5. Electrochemical Performances

All the EIS analysis was conducted with an IM6ex (Zahner-Elektrik GmbH & Co. KG instrument, Kronach, Germany) for measuring the ionic conductivity (σ) of as-prepared electrolytes in the temperature range from 30 to 80 °C at an interval of 10 °C, within the frequency range of 0.1 Hz to 10^5^ Hz at the open-circuit potential, and with an AC amplitude of 5 mV. The cells were allowed to reach thermal equilibrium for 40 min before each test. Then, the data of EIS spectra were fitted with an equivalent circuit model using Z-view software (version 3.1, Scribner Associates Inc., Southern Pines, NC, USA). The value of σ was calculated according to the following equation.
(1)$$\sigma =\frac{l}{AR}$$
where σ is the ionic conductivity, *l* is the distance between the two electrodes, *R* is the bulk resistance, and A is the active area of the electrode surface in contact with electrolytes.

The electrochemical measurements of as-prepared SPEs were assembled inside the glovebox under Ar-filled environment (H_2_O and O_2_ < 0.1 ppm) and carried out on an Ivium-n-Stat (Ivium Technologies, Eindhoven, The Netherlands). The electrochemical stability of SPEs was studied using LSV technology. All of the electrolytes were assembled based on Li/SPEs/LFP cells at room temperature and a constant rate of 1.0 mV/s from 0 V to 5 V. The *t*_Li+_ was also observed by the Swagelok cell (Appendix A). Then, the *t*_Li+_ of SPEs was calculated from the Bruce–Vincent–Evans Equation (2) [22,23].
*t*_Li+_ = (I_s_(∆U − I_0_R_0_))/(I_0_(∆U − I_s_R_s_))(2)
where I_0_, I_s_, R_0_, R_s_, and ∆U are initial current, steady-state current, interfacial resistance without and with polarization, and applied DC polarization voltage, respectively.

## 3. Results and Discussion

### 3.1. Chemical Structure Characterization

The fact that the color of electrolytes was untransparent and unstable (Appendix A) with low concentrations (1 M and 1.5 M) of LiFSI meant that the in situ polymerization was not successful; therefore, we selected SPE-2.5 to evaluate the polymerization condition.

The chemical structure of SPE-2.5 was confirmed by ^1^H-NMR and FTIR spectroscopy (as Figure 2a,c shows, respectively). The detailed ^1^H-NMR analytical data are listed as follows. ^1^H-NMR (DMSO-D_6_, ppm): δ = 0.81 (t, 3H), 1.26 (s, 2H), 1.61 (q, 2H), 3.15 (s, 2H), 3.26 (m, 2H), 3.5 (d, 2H), 4.2 (dd, 2H), 4.3 (dd, 2H), and 4.8 (t, 1H). The number-average molecular weight of poly (EOM) was estimated from the relative intensities of the NMR peaks between the terminal structure (3.5 ppm) and the main chains (3.24 ppm), which was 7640 [15]. However, the estimated molecular weights of SPE-2 and SPE-3 were ca. 4920 and ca. 6090, respectively, estimated from their NMR spectra (Appendix A). Poly (EOM)-based SPEs have two possible sites which coordinate with lithium ions. One is the terminal hydroxy group, which probably possesses the good solubility of Li salts; the other one is the trimethylene oxide moiety in the backbone [15].

Figure 2b exhibits the FTIR spectra of SPEs. For comparison, we also measured the FTIR spectra of the precursor EOM. As-prepared SPEs displayed the disappearance or reduction of the peak at ca. 960 cm^−1^ (Figure 2b,c), which can be attributed to the functional group -C-O-C [24] indicating the successful polymerization of poly (EOM). The broad peaks in the range of 3100–3700 cm^−1^ for all electrolytes indicate the presence of H-bonded N anions along with the trace amounts of physiosorbed water molecules [25]. SPEs also displayed the main peaks of -CH_3_ and -CH_2_ stretching at 2962 and 2882 cm^−1^, respectively. In addition, the peak at 840 cm^−1^ is ascribed to Li-O rocking, indicating the interaction of ether oxygen atoms with Li ion [24].

To further confirm polymerization, we investigated the conversion rate (CR) of the polymerized electrolyte, which was evaluated by measuring the peak area of oxetane ether bands (ca. 960 cm^−1^ or 840 cm^−1^) at each time point of the reaction, and determined using the following Equation (3) [10,26]:(3)$$\mathrm{Conversion}\;\mathrm{rate}=\frac{A_{0}-A_{t}}{A_{0}}\times 100\%$$
where CR is the conversion at time t, and *A*_0_ and *A_t_* are the peak areas of the functional groups before polymerization and at time t, respectively. Figure 3 shows the polymerization CR of synthesized SPEs, and the CR values of SPE-2, SPE-2.5, and SPE-3 reached ca. 60.52%, 65.42% and 72.46%, respectively. These results indicate the presence of SPEs, along with unreacted start materials, and that the suitable mole ratio of Li to O played a vital role in the polymerization process [27].

### 3.2. Physicochemical Properties of SPEs

Figure 4a shows the TGA plots of SPE-2, SPE-2.5, and SPE-3. All the SPEs exhibited a two-step weight loss. The weight loss (ca. 12%) up to 210 °C in the first step can be ascribed to the loss of physiosorbed water molecules and the incorporation of LiFSI, which is consistent with the reported work [25,28]. Then, all SPEs experienced a sharp weight loss up to 500 °C, and the weight losses of SPE-3, SPE-2.5, and SPE-2 were ca. 75%, 78%, and 82%, respectively. The residual weights (%) of SPE-2, SPE-2.5, and SPE-3 were ca. 9%, 14%, and 24%, respectively, up to 600 °C. This is probably ascribable to residual carbon compounds from their decomposition, which needs further in-depth observation. The residual percentage of SPE-3 was higher those of SPE-2 and SPE-2.5 because of the high concentration of lithium salt.

Figure 4b exhibits the viscosities of SPE-2 (1 M), SPE-2.5 (1 M), and SPE-3 (1 M) which were evaluated by dissolving them in DMSO. Their viscosities were ca. 2.17, ca. 2.46, and ca. 2.52 cP at 30 °C, respectively. Nevertheless, at 80 °C, their dynamic viscosities decreased to ca. 1.13, ca. 1.36, and ca. 1.42 cP, respectively. As the temperature increased, the dynamic viscosity of the electrolyte presented a decreasing trend, which can be ascribed to the weakening intermolecular forces in the electrolyte [25]. This low viscosity of all SPEs implies that they possess the capacity of conducting lithium ions with high conductivity.

### 3.3. Electrochemical Performances

#### 3.3.1. Ionic Conductivity and Impedimetric Stability

The conductivity was evaluated using EIS technology, and the σ values of SPE-2, SPE-2.5, and SPE-3 were calculated from Nyquist plots, fitting results with an equivalent circuit (Appendix A); the calculated values were 0.14 mS/cm, 0.25 mS/cm, and 0.16 mS/cm at 30 °C, respectively. The conductivities of SPE-2, SPE-2.5, and SPE-3 were 0.88 mS/cm, 1.3 mS/cm, and 1.1 mS/cm at 80 °C, respectively. These conductivity values were higher than reported poly (oxetane)-based electrolytes [9,10]. Moreover, the conductivity of SPE-2.5 was higher than those of SPE-2 and SPE-3, which indicated a suitable concentration of Li salt enhanced the conductivity of the electrolyte [10,23]. To study the temperature dependence σ of as-synthesized SPEs, we drew ln σ vs. the inverse of absolute temperatures, as displayed in Figure 5b. The plot indicates a linear dependency of ln σ along with temperature, which agrees with the typical Arrhenius plot. The activation energy (*E_a_*) was calculated from Arrhenius plots. The *E_a_* of SPE-2.5 (ca. 0.25 eV) was lower than those of SPE-2 and SPE-3, which were ca. 0.3 eV and ca. 0.32 eV, respectively. The low *E_a_* of electrolytes could be ascribed to the weak binding energies between the Li^+^ cation and the corresponding anions, and facilitates high ionic conductivity [29,30].

The successive EIS measurements were utilized to investigate the electrochemical stability of SPE-2, SPE-2.5, and SPE-3. The series of EIS measurements were as follows: 10 CV sweeps (potential range from −1.5 to 5 V, scan rate 25 mV/s); relaxation at 0 V (1 min). This series of electrochemical stability testing was repeated 11 times. Figure 5c displays the change of σ values along with the number of EIS scans. The σ values of all electrolytes were decreased after CV sweeping, and were ca. 22.85%, ca. 9.60%, 19.23% for SPE-2, SPE-2.5, and SPE-3, respectively. This indicates that these SPEs may develop the electrochemical stability of LiBs. Additionally, the FE-SEM image (Figure 5d) of SPE-2.5 displays good electrolyte/electrode contact, which also facilitates conductivity. Meanwhile, the SEM image also reveals that good compatibility of electrolytes and electrodes could enhance electrochemical stability (Appendix A) [19].

#### 3.3.2. Electrochemical Stability Window and *t*_Li+_ Measurement

Figure 6a displays the kinetic ESW of as-prepared SPEs. The oxidation potentials (vs. Li/Li^+^) of SPE-2, SPE-2.5, and SPE-3 were up to ca. 3.0, 3.75, and 3.4 V, respectively. Notably, the working potential of SPE-2.5 was a little higher than those of LiFePO_4_ (3.4 V), LiNi_1/3_Mn_1/3_Co_1/3_O_2_(NMC) (3.7 V), LiNi_0.8_Co_0.15_Al_0.05_O_2_(NCA) (3.7 V), and 0.5Li_2_MnO_3_·0.5LiMO_2_ (3.6 V), which are current commercially available cathodes [31]. The low operation potentials of SPE-2 and SPE-3 were due to the abundant hydroxy group coactions and the weak motion of lithium ions in the electrolyte.

The *t*_Li+_ of the electrolyte plays an important role in evaluating the electrochemical performance of LiBs. Based on the values of σ and ESW, we only studied the *t*_Li+_ of SPE-2.5. the homogeneous liquid electrolyte was injected into a Teflon chamber and sealed in a Swagelok cell in an Ar-filled environment glovebox and kept at 60 °C for 54 h After complete polymerization, the cell was cooled to ambient temperature. Measurements were carried out at 25 °C. Figure 6b displays the chronoamperometric traces of the symmetrical Swagelok cell at the applied voltage of 10 mV and the inset of Figure 6b exhibits the Nyquist plots of the cell before and after polarization. The details of the analytical parameters of chronoamperometric and Nyquist Plots are summarized in Appendix A, which obtained the *t*_Li+_ of ca. 0.74. The *t*_Li+_ of this electrolyte was much higher than what has been reported in the literature, as summarized in Appendix A. This indicates that Li^+^ is mainly carried for ionic charge in the SPE-2.5, rather than the counter anion [25].

#### 3.3.3. Electrochemical Performances

In general, higher ionic conductivity of electrolytes correlates well with better electrochemical performance. Thus, we investigated the battery performance of SPE-2.5 with a device configuration of LiFePO_4_/SPE-2.5/Li metal. Due to stable cycling performance, high theoretical capacity (170 mAh/g at 0.1 C), and a high tolerance to overcharge [22,25], LiFePO_4_ was used as a cathode material. The C-rate was calculated by cathode material weight (12.80 mg/cm^2^).

Figure 7a demonstrates the CD profiles of LiFePO_4_/SPE-2.5/Li at 1st, 25th, and 50th cycle numbers over the potential range of 2.5–4.2 V at 0.1 C. The discharge specific capacities of SPE-2.5 containing the half-cell after 1st, 25th, and 50th cycles were ca. 141, 129, and 120 mAh/g, respectively. At the same time, the corresponding charge specific capacities of this electrolyte were ca. 157, 137, and 125 mAh/g, respectively. Figure 7b exhibits the variation of C_sp_ and η with the number of CD cycles at 0.1 C. The C_sp_ of SPE-2.5 electrolyte-based half-cell was ca. 120 mAh/g at 0.1 C after 50 CD cycles, which retained 85.11% of initial C_sp_. This could be ascribed to the effect of LiFePO_4_ cathode material thickness (our LiFePO_4_ thickness was ca. 55 μm) during the in situ polymerization process [32]. In addition, the η of the half-cell was ca. 95.25% after 50 CD cycles, indicating that there was a highly reversible positive electrochemical reaction. Over 2 M LiFSI in electrolytes probably generates solid electrolyte layers, which facilitate cycling performances of Li metal anode [33]. Thereof, these results suggest that SPE-2.5 electrolyte is promising for the development of Li-ion batteries.

#### 3.3.4. Mechanism of In Situ Cationic Ring Opening of Poly (EOM)

According to our experimental results, a plausible self-catalyzed cationic ring-opening polymerization mechanism (Figure 8a) and merits of in situ polymerization (Figure 8b) were presented. Due to the absence of an inflammable solvent, as-prepared SPEs could improve the safety performance of batteries. Ionic conductivity was observed to be better in SPE-2.5 than in SPE-2 and SPE-3, suggesting that abundant OH groups and highly concentrated Li salt affected the Li+ ion mobility in poly (EOM). Additionally, SPE-2.5 demonstrated a comparable C_sp_ and η, which could be applied in Li-ion batteries.

## 4. Conclusions

Herein, we successfully synthesized a solid polymer electrolyte based on poly (EOM) via an in situ polymerization strategy with LiFSI as an initiator. The in situ polymerization approach greatly simplified the fabrication process of polymer electrolytes for LiBs. Furthermore, the in situ-constructed framework of SPEs remarkably enhanced the contact compatibility and interfacial stability between the SPEs and the electrodes. Additionally, we obtained SPEs which exhibited considerable conductivity owing to the absence of organic solvent and the compatibility of LiFSI with anode. The as-prepared SPE-2.5 possessed comparable conductivity (0.25 mS/cm at 30 °C and 1.3 mS/cm at 80 °C), oxidation stability up to 3.75 V, high *t*_Li+_ (0.74), and cycling performance after 50 charge–discharge cycles. These results suggest a strategy which favors the consideration of the in situ-formed poly (EOM)-based polymer electrolyte as a promising candidate for lithium-metal rechargeable batteries.

## Figures and Tables

**Figure 1 membranes-12-00330-f001:**
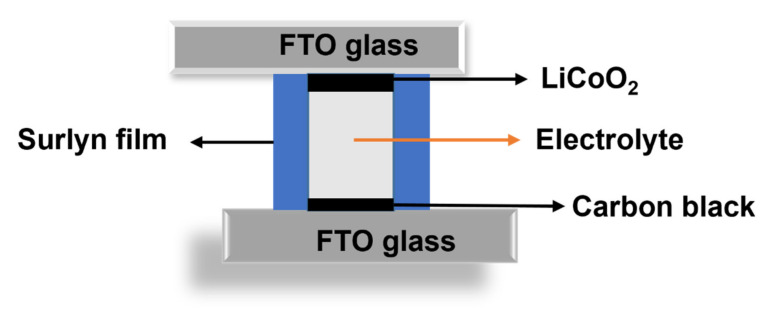
Schematic diagram of an asymmetric dummy cell.

**Figure 2 membranes-12-00330-f002:**
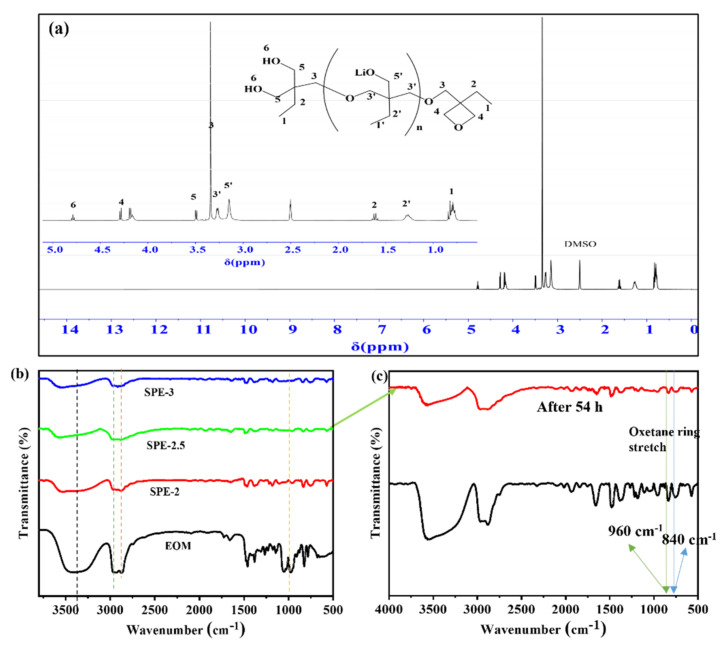
(**a**) ^1^H-NMR spectra (inset shows the partial enlarged ^1^H-NMR spectra) of SPE-2.5, (**b**) FTIR spectra of precursor EOM and SPEs and (**c**) initial transmittance and after 54 h of SPE-2.5.

**Figure 3 membranes-12-00330-f003:**
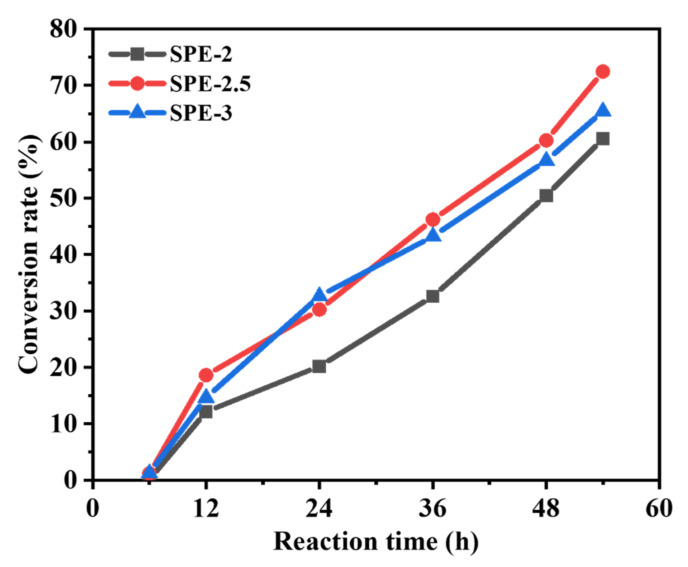
Conversion rate vs. reaction time plots of SPE-2, SPE-2.5, and SPE-3.

**Figure 4 membranes-12-00330-f004:**
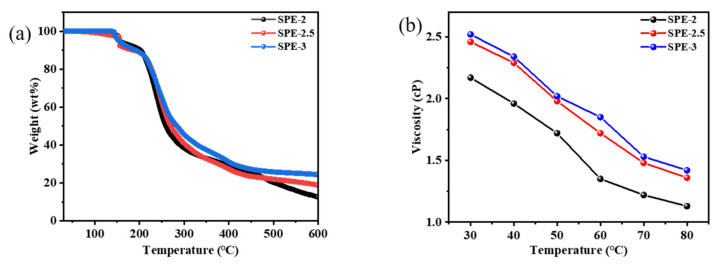
TGA traces (**a**) and viscosity vs. temperature curves (**b**) of SPE-2, SPE-2.5, and SPE-3.

**Figure 5 membranes-12-00330-f005:**
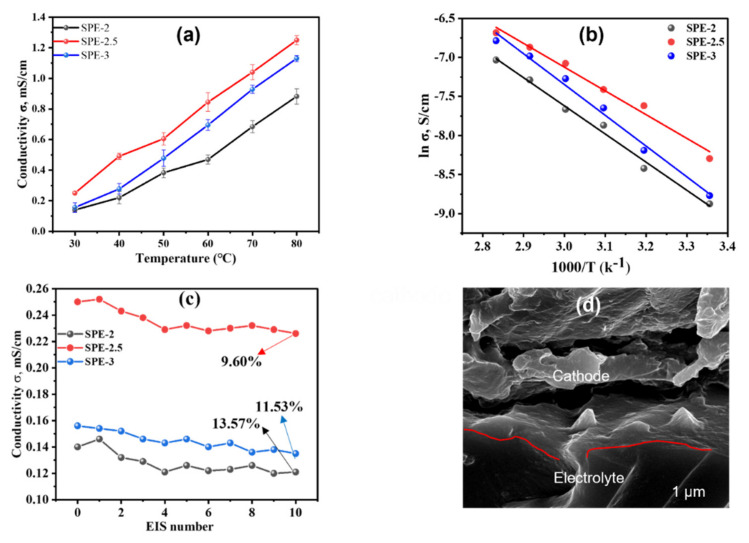
Plots of conductivity vs. temperature (**a**) and ln (Li^+^ conductivity) vs. the inverse of absolute temperature (**b**) of SPEs. (**c**) Variation of the conductivity of SPEs with repetitive EIS scanning for SPE-2, SPE-2.5, and SPE-3, at 25 °C and open circuit condition. (**d**) Cross-sectional FE-SEM image after 50 cycles from dummy cell of SPE-2.5.

**Figure 6 membranes-12-00330-f006:**
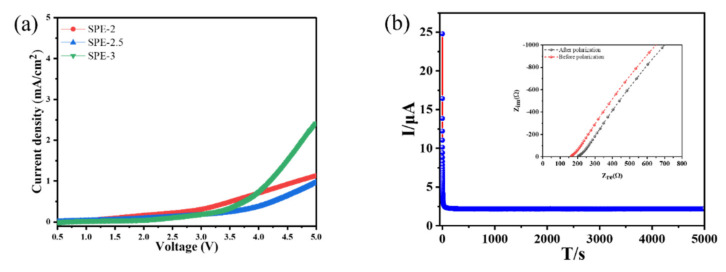
(**a**) Current density vs. voltage plot of SPE-2, SPE-2.5, and SPE-3. (**b**) Chronoamperometric polarization plot of the electrolyte-based symmetry Li/SPE-2.5/Li Swagelok cell at 10 mV for 5000 s (inset exhibits the Nyquist curves of the cell before and after polarization).

**Figure 7 membranes-12-00330-f007:**
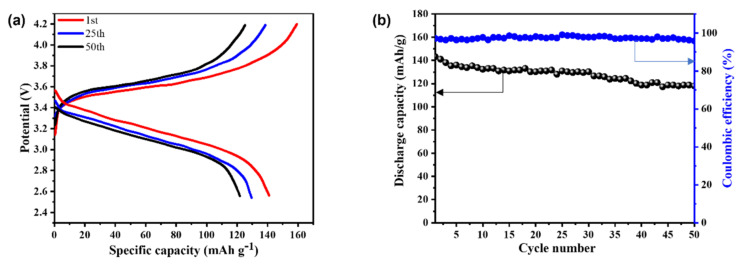
CD plots of SPE-2.5 with structure of LiFePO4/SPE-2.5/Li cell for 0.1 C over a potential range of 2.5–4.2 V at 25 °C (**a**), discharge capacity and coulombic efficiency of CD cycling number of the LiFePO4/SPE-2.5/Li cell at 0.1 C and 25 °C (**b**).

**Figure 8 membranes-12-00330-f008:**
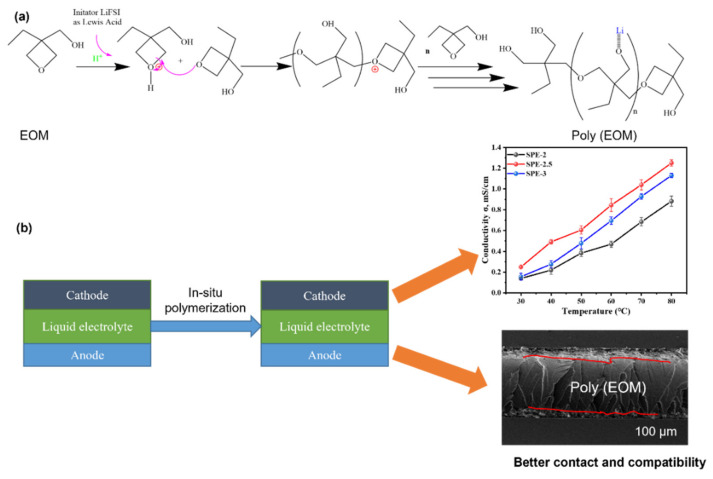
(**a**) Schematic illustration of cationic ring-opening polymerization of poly (EOM), (**b**) advantages of in situ polymerization.

## Data Availability

The data presented in this study are available on request from the corresponding author.

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
