# Peer review of "Lithium Salt Catalyzed Ring-Opening Polymerized Solid-State Electrolyte with Comparable Ionic Conductivity and Better Interface Compatibility for Li-Ion Batteries"

_membranes, 2022, doi:10.3390/membranes12030330_

Round 1

Reviewer 1 Report

This paper on synthesis methods of polymers for lithium-ion battery is of great interest for the community.

The paper is well written. However, a sentence not complete in page 9.

I suggest the editor to accept this paper with minor corrections, i.e.:

- the units of the number-averaged molecular weight should be precise (p4)

- SPE 2 and 3 should be analyzed by NMR and molecular weights also given

Reviewer 2 Report

The manuscript "Lithium Salt Catalyzed Ring-opening Polymerized Electrolyte with Comparable Ionic Conductivity and Better Interface Compatibility for Li Cell" shows an interesting concept containing SPE for Li batteries with sound results.

The authors used many abbreviations so maybe it might be helpful define those in a Table in supplementary. The synthesis are well performed as well the analytic measurements of the assembling.

A general question why is it advantage using Poly (EOM) SPE and why did the in-situ polymerization didn't work for other SPE than SPE2.5? Do the authors have an explanation and maybe an idea how those can be made successfully?

In case of Li-batteries on commercial application the toxicity of those must be as well evaluated. Did the authors investigate such or have some citation of others made it in similar way? 

Beside the intensive characterizations the authors provide, did you made some cyclic voltammetric measurements and it would be interesting to show (can be supplementary) also included the coulo-voltammetric cycle (charge density potential curve. In general if there is thermodynamic and electrochemical balance (closed loop of charge density curve) there are no irreversible effects taking place during charging/discharging cycles. 

Regarding Li-Batteries, did the authors perform durability and long-term cycle stability of those devices? It would be good to add those (can be as well supplementary). So far the obtained date either ion conductivity didn't show any standard deviation. Did the authors made only one samples for each SPE? If not please use mean values with standard deviation.

If comparing the different SPE why the 2.5 has the best performance. Please explain it in more details in the result part as well conclusion.

Reviewer 3 Report

This manuscript entitled "Lithium Salt Catalyzed Ring-opening Polymerized Electrolyte with Comparable Ionic Conductivity and Better Interface Compatibility for Li Cell" has been reviewed. In the paper, authors reported a solid-state polymer electrolyte based on LiFSI and EOM via in-situ thermal induced polymerization technique. The goal of the paper is good, however, the work in its present form is not complete due to lack of necessary battery cycle test. Therefore, I suggest accepting this work for publication after major revision. There are some issues need to be resolved before:

  1. It is better to use “lithium-ion battery” instead of “Li Cell” in Tittle. Moreover, the keyword “solid-state” should be mentioned in Tittle.
  2. For EIS, if the bulk resistance is obtained via fitting Nyquist plots with an equivalent circuit, please provide the corresponding fitting accuracy.
  3. As-prepared electrolytes show well ionic conductivity, so I hope to see their battery performance. In Figure 5c, the change of σ values along with the number of charge-discharge cycles is suggested.
  4. For Figure 5d, I can not observe solid electrolyte interphase layer in present magnification. Please provide clearer SEM image.

Round 2

Reviewer 3 Report

The revised manuscript has made proper revisions, so I recommend it be accepted for publication.